# Immune Effects of the Nitrated Food Allergen Beta-Lactoglobulin in an Experimental Food Allergy Model

**DOI:** 10.3390/nu11102463

**Published:** 2019-10-15

**Authors:** Anna S. Ondracek, Denise Heiden, Gertie J. Oostingh, Elisabeth Fuerst, Judit Fazekas-Singer, Cornelia Bergmayr, Johanna Rohrhofer, Erika Jensen-Jarolim, Albert Duschl, Eva Untersmayr

**Affiliations:** 1Institute of Pathophysiology and Allergy Research, Center for Pathophysiology, Infectiology and Immunology, Medical University of Vienna, 1090 Vienna, Austria; anna.ondracek@meduniwien.ac.at (A.S.O.); denise.heiden@meduniwien.ac.at (D.H.); elisabeth.stafflinger.fuerst@gmail.com (E.F.); judit.singer@ist.ac.at (J.F.-S.); cornelia.bergmayr@gmail.com (C.B.); johanna.rohrhofer@hotmail.com (J.R.); erika.jensen-jarolim@meduniwien.ac.at (E.J.-J.); 2Biomedical Sciences, Salzburg University of Applied Sciences, 5412 Puch/Salzburg, Austria; geja.oostingh@fh-salzburg.ac.at; 3The interuniversity Messerli Research Institute of the University of Veterinary Medicine Vienna, Medical University of Vienna, and University of Vienna, 1210 Vienna, Austria; 4Department of Biosciences, University of Salzburg, 5020 Salzburg, Austria; albert.duschl@sbg.ac.at

**Keywords:** beta-lactoglobulin, food allergy, nitration

## Abstract

Food proteins may get nitrated by various exogenous or endogenous mechanisms. As individuals might get recurrently exposed to nitrated proteins via daily diet, we aimed to investigate the effect of repeatedly ingested nitrated food proteins on the subsequent immune response in non-allergic and allergic mice using the milk allergen beta-lactoglobulin (BLG) as model food protein in a mouse model. Evaluating the presence of nitrated proteins in food, we could detect 3-nitrotyrosine (3-NT) in extracts of different foods and in stomach content extracts of non-allergic mice under physiological conditions. Chemically nitrated BLG (BLGn) exhibited enhanced susceptibility to degradation in simulated gastric fluid experiments compared to untreated BLG (BLGu). Gavage of BLGn to non-allergic animals increased interferon-γ and interleukin-10 release of stimulated spleen cells and led to the formation of BLG-specific serum IgA. Allergic mice receiving three oral gavages of BLGn had higher levels of mouse mast cell protease-1 (mMCP-1) compared to allergic mice receiving BLGu. Regardless of the preceding immune status, non-allergic or allergic, repeatedly ingested nitrated food proteins seem to considerably influence the subsequent immune response.

## 1. Introduction

Food consists of a mixture of different proteins, carbohydrates, and fatty acids constituting a complex matrix susceptible to modifications [1]. Any potential food allergen in the matrix could undergo a range of chemical alterations influencing gastrointestinal digestion, absorption, and presentation to the immune system. Modifications frequently investigated in recent research are nitration, oxidation, reduction, glycation, aggregation, and cross-linking [2,3,4,5,6].

Nitration is a chemical reaction targeting the aromatic ring of amino acid residues such as tyrosine and tryptophan [7]. Radical formation enables tyrosine to directly interact with nitrogen dioxide (NO_2_), or with free radical nitric oxide via the intermediate 3-nitrosotyrosine, to form 3-nitrotyrosine (3-NT) [8].

The non-enzymatic nature of nitration allows the modification of airborne primary biological aerosol particles, e.g., pollens, which can be explained by their exposure to ozone and subsequent reaction with NO_2_ [9]. Allergens from fungal spores that have been exposed to ambient pollutants of urbane air were found nitrated and showed increased IgE binding capacity [10]. Traffic-related air pollution can thus promote nitration of airborne allergens, an effect that could potentially extend to crops and grains. Moreover, the presence of 3-NT was demonstrated in proteins of meat products [11,12,13,14] and was connected to the addition of curing agents such as nitrite in the course of food processing and storage [11,12], explaining recent interest to establish improved methods for the detection of 3-NT in food [15].

Nitration is not limited to exogenous processes but has also been described endogenously during inflammation, in neurodegenerative diseases, or aging [16,17]. Most importantly for food proteins, the intragastric milieu supports the formation of NO_2_ derived from dietary nitrite contained in green leafy vegetables [18].

According to current literature, nitrated allergens should be regarded as a major concern for allergic individuals. They were shown to trigger enhanced mediator release in rat basophil leukemia cell assays and acted as a more potent T cell stimulus [4,19]. Modified tyrosine residues are reported to change recognition of antigens by immune cells, leading to an altered immune response [20]. Recently, we reported that systemic challenge with nitrated beta-lactoglobulin (BLGn) in pre-existing food allergy aggravated the anaphylactic response as measured by the anaphylaxis marker mouse mast cell protease-1 (mMCP-1) and core body temperature compared to untreated BLG (BLGu) [21].

However, the bioavailability of allergens in the intestine is regulated by gastrointestinal digestion [22,23]. Considering that individuals might get repeatedly exposed to nitrated proteins via daily diet, we addressed in this study whether the presence of 3-NT could influence the immune system after repeated exposure via the gastrointestinal tract. As effector pathways of the immune system might be differently affected in non-allergic or allergic organisms, we performed experiments on a non-allergic and an allergic immune background, using our established food allergy protocol [24] and the model allergen BLG in nitrated (BLGn) and untreated (BLGu) form.

## 2. Materials and Methods

### 2.1. Protein Extraction from Food and Stomach Content

Proteins from soy, hazelnut, walnut, celery, and wheat were isolated by water extraction. Twenty grams of source food material was crushed, mixed with 50 mL of distilled water, and mixed for 2 h using a magnetic stirring rod. Subsequently, solutions were centrifuged at 3000× *g*, filtrated, and lyophilized.

Stomach contents collected by gastric lavage from six non-allergic, six-week-old female BALB/cAnNCrl mice were mixed with 50 µL distilled water and shaken for 6 h at room temperature (RT). Afterwards, samples were centrifuged at 10,000× *g* and protein concentrations of supernatants were determined using the Pierce BCA protein assay kit (Thermo Scientific, Vienna, Austria).

### 2.2. Dot Blot Experiments

Samples were analyzed in triplicates by dotting 1 µL of gastric lavage (14 mg/mL) or food extract solution (1 mg/mL for soy, hazelnut, walnut, and celery, and 0.33 mg/mL for wheat) on a nitrocellulose blotting membrane (0.2 µm from GE Healthcare Life sciences, Vienna, Austria). For control purposes, a chemically nitrated protein, an untreated protein, and mouse chow extract were included in the assay.

After drying, unspecific binding sites were blocked with 5% goat serum in tris-buffered saline supplemented with 0.1% Tween 20 (Sigma, Vienna, Austria; 0.1% TBST), then membranes were incubated with rabbit anti-3-NT antibody (1:5000, Merck Millipore, Vienna, Austria) dissolved in 0.1% TBST containing 5% BSA (dilution buffer) at 4 °C overnight. A second control membrane was incubated indilution buffer. Horseradish peroxidase-labeled goat anti-rabbit IgG (1:5000 in dilution buffer (Thermo Scientific, Vienna, Austria) was added to both nitrocellulose membranes. Membranes were incubated with a substrate (Super Signal West Pico Chemiluminescent substrate, Thermo Scientific, Vienna, Austria) and developed after 15 min of exposure.

### 2.3. Nitration of BLG

Protein nitration was performed using tetranitromethane (TNM, Sigma, Vienna, Austria) as a nitrating agent as described previously [19]. One mg/mL BLG (Sigma, Vienna, Austria) dissolved in Na_2_HPO_4_ buffer (10 mmol/L, pH 7.4) was mixed with 0.5 mol/L TNM in methanol at a TNM/tyrosine molar ratio of 15/1 under continuous agitation for 60 min in glass bottles. Samples were washed three times with Na_2_HPO_4_ buffer using an Amicon Ultra-15 centrifugal filter device (Merck Millipore) with a 10 kDa cut-off membrane for eight minutes at 2860× *g* and the resulting protein concentrations were determined with the Pierce BCA protein assay kit (Thermo Scientific, Vienna, Austria) using BLG for standard curve preparation. To determine the number of nitrated tyrosine residues per molecule, 3-NT (Sigma, Vienna, Austria) was dissolved in 0.05 mol/L NaOH for the preparation of a standard curve ranging from 6.125 µmol/L to 200 µmol/L. Protein samples were diluted 1:2 in 0.1 mol/L NaOH. Absorbance was measured at 428–650 nm (TECAN, infinite M200 PRO), and the number of 3-NT per molecule was calculated dividing the molar concentration of 3-NT by the molar concentration of BLG.

### 2.4. In Vitro Digestion Experiments

Gastric digestion was simulated using a pharmaceutical gastric enzyme pill containing pepsin (Enzynorm, AstraZeneca) dissolved in 50 mL of 0.15 mol/L sodium chloride solution. Equal amounts of BLGu and BLGn (1 mg/mL, respectively) were incubated at a ratio of 1:1.25 mg/mL with simulated gastric fluid at pH 1.5 for 120 min at 37 °C on a shaker. The reaction was stopped by the addition of 1 mol/L sodium hydroxide solution. Digested and undigested samples were then compared by SDS-PAGE using Coomassie brilliant blue staining.

### 2.5. Animals

Six to eight week old female BALB/cAnNCrl mice (*n* = 30, 15–20 g, provided with a health report certificate) were purchased from the Core Facility for Biomedical Research, Division for Laboratory Animal Science and Genetics (Himberg, Austria) and housed under conventional conditions (circadian rhythm of 12 h light and dark cycles at 22 °C). Mice were kept in groups of six at the animal facility of the Institute of Pathophysiology and Allergy Research in polycarbonate Makrolon type II cages (Ehret GmbH, Emmendingen, Germany) with aspen wood bedding (Ehret GmbH, Emmendingen, Germany) sealed with filter tops. Cages were enriched with red transparent plastic nest boxes and nesting material of cellulose. Mice had ad libitum access to food (egg and cow’s milk-free diet, Ssniff, Soest, Germany) and water, except on days of oral gavages, when they were fasted 4 h prior to the experiment. Experimental procedures were carried out after a two-week acclimation period per group in a separate animal experimentation room in random order (within each group). Study design complied with the concept of the 3Rs (replacement, refinement, and reduction). Sample sizes were calculated according to previously published experiments [24] with a power calculation based on a two-sided two-sample *t*-test. The primary outcome of the in vivo experiments was the analysis of the type of immune response elicited by BLGn via the oral route compared to BLGu. Sentinel animals (*n* = 2) included in sensitization and experimental procedures were screened for their health status, with a special focus on bacterial infections, such as *H. pylori*, following the FELASA criteria [25]. Mice were treated according to the European Union guidelines of animal care and with approval of the animal ethics committee of the Medical University of Vienna and the Austrian Federal Ministry of Science and Research (approval number: BMWF-66.009/0270-II/3b/2013).

A schematic overview of the mouse experiments is given in Appendix A. Clinical symptoms after treatments were assessed by a blinded investigator following a previously published scoring system [26]: “0 = no symptoms; 1 = scratching and rubbing around the nose and head; 2 = puffiness around the eyes and mouth, pilar erecti, reduced activity, and/or decreased activity with increased respiratory rate; 3 = wheezing, labored respiration, and cyanosis around the mouth and the tail; 4 = no activity after prodding or tremor and convulsion; and 5 = death.”

### 2.6. Oral Gavages to Non-Allergic Mice

Two groups of non-allergic Balb/cAnNCrl mice (*n* = 6) were fed intragastrically (ig.) three times in weekly intervals using an animal feeding needle with 2 mg BLGu and BLGn, respectively (Appendix A). Blood was taken before and after oral gavages by puncture of the facial vein, and on day of sacrifice by heart puncture.

### 2.7. Sensitization Protocol and Treatment of Allergic Mice

Six-week-old female Balb/cAnNCrl mice (*n* = 12) were immunized as previously described by oral treatment cycles (Appendix A) [24]. In brief, mice were injected intravenously with 116 µg of the proton pump inhibitor (PPI) omeprazole dissolved in 0.9% sodium chloride on days 1–3, 15–17, 29–31, and 43–45 for gastric acid suppression. On days 2 and 3, 16 and 17, 30 and 31, and 44 and 45, mice were fed ig. with 200 µg BLGu mixed with 2 mg of the antacid sucrose sulfate-aluminum complex sucralfate in PBS 15 min after a second PPI injection. On days 60, 71, and 92, animals were injected intraperitoneally (ip.) with 2 µg BLGu absorbed to 2% aluminum hydroxide solution to induce a sustained systemic immune response. After immunization, animals were divided into two groups (*n* = 6, each) for ig. gavages. On days 114, 121, and 128 mice were gavaged ig. with 2 mg BLGu or BLGn (Appendix A). Blood was collected before and after immunization and 1 h after the last oral gavage of BLGu or BLGn by puncture of the facial vein, and on day of sacrifice by heart puncture.

### 2.8. Measurement of Cytokine Release from Spleen Cells

Spleens of six naïve, six-week-old female Balb/cAnNCrl mice that were not treated and of all 24 animals included in the treatment protocol were removed under sterile conditions, minced and passed through sterile nylon cell strainers (40 µm). Erythrocytes were lysed for five minutes using 5 mL ACK lysing buffer (Lonza, Basel, Switzerland). Cells were re-suspended, washed twice, and counted using an automated Coulter Counter (TC10; Bio-Rad). Mesenteric lymph nodes were pooled per experimental group and treated as described for spleens. Cells (4 × 10^5^ cells/well) were incubated in quadruplicates with BLGu (2.5 µg/mL), BLGn (2.5 µg/mL) based on results from previous titration experiments (data not shown), and concanavalin A (2.5 µg/mL, positive control, Sigma, Vienna, Austria), and medium as negative control, respectively, for 72 h at 37 °C. Supernatants were screened for IL-4, IL-10, and IFN-γ concentrations by ELISA following the manufacturer’s instructions (eBioscience, Vienna, Austria). Microtiter plates were coated with capture antibody overnight at 4 °C and blocked for 1 h at RT. Samples and standards were incubated overnight at 4 °C. After the addition of detection antibody and detection by avidin-horseradish peroxidase, plates were incubated with tetramethylbenzidine (TMB, BD Biosciences, Heidelberg, Germany). The reaction was quenched with 1.8 mol/L H_2_SO_4_. Optical density (OD) was measured at 450 and 570 nm as reference. Concentrations were determined by calculating OD values against the appropriate standard curves. Baseline levels were subtracted.

### 2.9. Flow Cytometry Analysis of Regulatory T Cells

Spleen cells and mesenteric lymph nodes were screened for Tregs by FC using a commercially available staining kit (eBioscience, Vienna, Austria) immediately after sacrifice. 1 × 10^6^ cells/sample were washed twice with flow cytometry (FC) buffer consisting of 1% fetal bovine serum (Gibco Invitrogen, Vienna, Austria) in PBS. Surface staining was performed by adding anti-CD4-FITC (1:400) and anti-CD25-APC (1:333) to each tube. Controls were treated equally using the appropriate isotype controls (rat IgG2a, 1:400 and rat IgG1, 1:333 in FC buffer, all from eBioscience, Vienna, Austria). Samples were incubated for 30 min at RT in the dark and then washed twice as described above. Cells were incubated in 500 µL Fix/Perm buffer for 30 min at 4 °C. Then, cells were washed using Perm buffer. Intracellular staining was performed using FoxP3-PE (1:40 in Perm buffer). Rat IgG2a-PE (1:40 in Perm buffer) was added to the isotype controls (eBioscience, Vienna, Austria). All samples were incubated for 30 min at 4 °C and then washed twice with Perm buffer. Cells were fixed using 300 µL fixation solutions. Acquisition was done using a FACS Calibur flow cytometer (BD Biosciences). Cells were separated from debris by forward and side scatter characteristics, and the stopping gate was set to 15,000 events. Gates were adjusted according to group-specific isotype controls. Tregs were identified from CD4^+^ CD25^+^ lymphocytes according to positive staining for FoxP3. The gating strategy is depicted in Appendix A. Analysis was done with BD FACSDiva Software (BD Biosciences).

### 2.10. Detection of Allergen-Specific IgE, IgA, IgG1, and IgG2a in Serum

Mouse serum samples were screened for BLG-specific IgA, IgE, IgG1, and IgG2a. Standards and BLG (10 µg/mL) were coated overnight at 4 °C in sodium carbonate coating buffer and then blocked with 1% bovine serum albumin (BSA, Sigma, Vienna, Austria) in PBS. Serum samples were diluted 1:20 for IgE antibody detection, 1:100 for IgG1 and IgG2a, and 1:200 for IgA in 0.1% BSA in TBS (dilution buffer). Rat anti-mouse IgA, IgE, IgG and IgG2a (BD Biosciences, Heidelberg, Germany) were added at a concentration of 1:500 in dilution buffer followed by a peroxidase-labeled goat anti-rat IgG (1:1000 in dilution buffer, Amersham, Buckinghamshire, UK). Bound antibodies were detected by TMB and the reaction was quenched with 1.8 mol/L H_2_SO_4._ The color reaction was measured at 450 nm (with 630 nm as a reference wavelength). Antibody titers were calculated according to the respective standard dilution series.

### 2.11. β-hexosaminidase Release Assay from Rat Basophil Leukemia (RBL-2H3) Cells

To assess the capacity of BLG-specific IgE to elicit mediator release upon allergen encounter [27], 4 × 10^4^ RBL-cells per well (a kind gift by A. Hartl) were incubated overnight and then passively sensitized with 5% serum of allergic mice for two hours at 37 °C. BLGu (10 µg/mL) was added to crosslink murine IgE bound to FcεRI for 30 min. Mediator release was determined indirectly by measuring the activity of β-hexosaminidase after the addition of the substrate 4-methylumbelliferyl-*N*-acetyl-β-d-galactosaminide (Sigma, Vienna, Austria). The reaction was stopped after one hour using glycine buffer (pH 10.7), and fluorescence of the formed product was measured at an excitation wavelength of 360 nm and an emission wavelength of 465 nm. Results are expressed relative to the positive control Triton X (Sigma, Vienna, Austria).

### 2.12. Detection of Allergen-Specific IgE and IgA in Intestinal Lavages

To determine local antibody production, intestines were flushed with 2 mL protease inhibitor (Roche, Vienna, Austria) dissolved in PBS and shaken for 4 h at 4 °C before centrifugation at 10,000× *g*. Supernatants were screened for BLG-specific IgA and IgE. Standards and BLG were coated overnight at 4 °C in coating buffer and then blocked with 1% BSA in PBS. Lavages were added undiluted to the respective wells. Rat anti-mouse IgE (1:500 in dilution buffer) and biotin anti-mouse IgA (1:250 in dilution buffer) incubation was followed by peroxidase-labeled goat anti-rat IgG (1:1000 in dilution buffer) or streptavidin-linked horseradish peroxidase. Bound antibodies were detected as described for serum ELISA. Antibody titers were calculated against the appropriate standard dilution series.

### 2.13. Detection of mMCP-1

Mouse MCP-1 was measured by ELISA according to manufacturer’s instructions (eBioscience, Vienna, Austria). Briefly, plates were coated with capture antibody overnight and blocked at RT for 1 h. Serum samples were diluted 1:30 in assay dilution reagent and incubated together with standards at 4 °C overnight. After incubation of detection antibody and avidin-horseradish peroxidase, TMB was used as substrate and quantified as described for cytokine ELISAs. Mouse MCP-1 levels were calculated according to a standard curve. Baseline levels were subtracted.

### 2.14. Statistical Analysis

All available samples were analyzed in groups as defined before experimental procedures. Data were statistically compared and visualized using IBM SPSS Statistics version 25 and GraphPad Prism version 5.00 for Windows (GraphPad Software, www.graphpad.com). Groups were checked for normal distribution by Kolmogorov–Smirnov normality test to ensure appropriate statistical comparison between two groups. The exact tests used are indicated in the figure legends. Data are presented as boxplots with whiskers defined according to Tukey, leaving outliers as black dots, or as scatter blots linking repeated measures of a single mouse.

## 3. Results

### 3.1. Nitrated Proteins Are Present in Food and Stomach Content Extracts of Naïve Mice

Protein extracts of soy, hazelnut, celery, wheat, and walnut were tested for the presence of nitrated residues in dot blot experiments using a 3-NT-specific antibody (Figure 1A). Soy, hazelnut, celery, and wheat were tested positive for 3-NT to varying degrees, with wheat showing the highest intensity despite the lower protein concentration. Walnut extract was negative for 3-NT.

The stomach contents of naïve mice and extracts of mouse feed were similarly tested. In five of six mice, the dot blot revealed naturally nitrated proteins in stomach contents. Samples of one mouse (M2) remained negative for 3-NT. The most prominent signal was seen in extracts of mouse chow, which, however, also displayed unspecific binding of the secondary antibody (Appendix A). Nitrated control proteins developed a strong signal, whereas untreated control proteins remained negative.

### 3.2. Nitration Decreased Stability of BLG in Simulated Gastric Fluid

To evaluate the impact of gastric enzymatic digestion on protein stability, BLGu and BLGn were analyzed in simulated gastric fluid experiments (Figure 2). BLGu remained stable for 120 min and was comparable to undigested control samples (Figure 2, left panel). In the presence of nitrated tyrosine residues, monomeric (18 kDa) as well as dimeric (36 kDa) BLG was rapidly degraded (Figure 2, right panel).

### 3.3. In Non-allergic Animals, BLGn Induced IgA, IL-10, and IFN-γ

The innate immunostimulatory capacity of BLGn was investigated stimulating spleen cells of six naïve, untreated mice (Table 1). While exposure to BLGn led to significantly higher IFN-γ release compared to BLGu, no differences could be detected for IL-10 and IL-4.

To evaluate the influence of orally applied nitrated food allergens on the immune response, non-allergic mice were fed three times in weekly intervals with BLGu or BLGn (Appendix A). Specific antibody levels in serum samples were compared before and after oral gavages. A significant elevation of specific IgA was detected after gavage of BLGu, but not of BLGn (Figure 3C). No induction of specific IgE could be observed, neither in response to BLGu nor to BLGn (Figure 3B). Accordingly, blood collected 1 h after the last gavage did not contain relevant concentrations of the anaphylaxis marker mMCP-1 in any group (Figure 3A). Moreover, no differences were observed for BLG-specific IgG1 or IgG2a levels, irrespective of time point, and BLG preparation (data not shown).

In parallel, intestinal lavages were screened for specific mucosal antibody titers. Specific IgE levels in lavages were below the detection limit (data not shown). BLG-specific IgA in lavages did not differ between experimental groups (Figure 3D). Screening for total IgA revealed comparable levels in both groups (Figure 3E).

To evaluate the cytokine secretion of splenocytes from both experimental groups, supernatants of cultured spleen cells were screened for IFN-γ, IL-10, and IL-4 production after stimulation with BLGu (Figure 4). Significantly higher levels of IFN-γ and IL-10 were detected in the BLGn group compared to the group fed with BLGu. For IL-4, no differences could be observed. Stimulation with BLGn (Appendix A) did not induce differences in IFN-γ, IL-10 and IL-4 release comparing the two groups of interest. Cytokine concentrations released from pooled, stimulated lymph node cells remained below the detection limits of the respective assays. Flow cytometry analysis showed that the Treg compartment remained unaffected by any of the treatments (Table 2).

### 3.4. In BLGu-Allergic Mice, Ingestion of BLGn Led to Significantly Altered Antibody Levels

The influence of ingested BLGn was next investigated in BLG-allergic mice pre-sensitized with BLGu. When serum from BLGu-allergic mice was used to passively sensitize IgE receptors on RBL-2H3 cells in vitro, mediator release could be elicited in response to BLGu (Figure 5A), thus confirming successful immunization. Specific antibody levels did not differ between groups before the start of the allergen gavages. Subsequently, groups were fed three times in weekly intervals with BLGu or with BLGn (Appendix A).

After three weeks, mMCP-1 levels determined in blood drawn one hour after the last gavage were moderately elevated in mice receiving BLGn but differed significantly from mice fed with BLGu (Figure 5B). We did not observe any clinical signs of anaphylactic responses nor a drop of core body temperature upon this last oral allergen exposure (data not shown).

No changes of specific IgE, IgA, and IgG2a were elicited by gavages in any of the groups (Figure 5C,D,G). Measuring specific IgG1 showed a uniform increase in the BLGu group (Figure 5E) whereas five of six mice of group BLGn had lower levels than before. The change of specific IgG1 before and after treatment induced by BLGu or BLGn diverged significantly (Figure 5F). Interestingly, allergic mice receiving BLGn via the oral route showed significantly higher titers of BLG-specific IgA in intestinal lavages compared to mice receiving BLGu (Figure 5H). BLG-specific IgE in intestinal fluids remained below the detection limit (data not shown).

BLGu-stimulation of spleen cells isolated from BLG-allergic animals after repeated oral exposure to BLGu or BLGn was not associated with significant differences regarding cytokine production (Figure 6, Appendix A). Flow cytometry analysis of T cells did not reveal significant differences regarding absolute counts or percentages of Tregs (Tregs/CD4^+^ cells) in response to weekly gavages (Table 2).

## 4. Discussion

Nitration of proteins is known to occur endogenously under oxidative and nitrosative stress [28] arising in inflamed tissues [29,30], the gastrointestinal tract [18], or the brain [16,31,32]. We discovered here that nitrated residues do occur in the stomach content of naive, untreated mice under physiological conditions. It has been hypothesized that nitrous acid derived from nitrite in the acidic environment of the stomach has the potential to initiate the formation of diverse nitrating species [5]. These endogenous nitration processes could chemically alter dietary or mucosal compounds in the digestive apparatus, as shown for pepsin and occludin [33,34].

However, the presence of 3-NT in the stomach content could also be attributed to exogenous nitration of food proteins. For instance, by environmental pollution, proteins can be nitrated upon reaction with radicals [9,35,36]. As this has just been described in airborne proteins so far, i.e., pollen allergens [10,37,38], we have tested its occurrence in food products and crops by dot blots of extracts. Of five tested, frequently consumed food products, four harbored 3-NT residues in detectable amounts. Wheat, which is one of the major nutritional food sources worldwide with millions of hectares of harvested crop [39], showed the most striking positive signal despite the lower protein concentration used. Interestingly, we have also found 3-NT in soy, which we have previously reported to promote the development of food allergies in our mouse model when soy is contained in the mouse chow [40].

Consequently, nitrated proteins can be present in our daily food, irrespective of the food group. We explored whether the immune response would be affected by constant exposure to 3-NT in non-allergic or allergic animals. To specifically determine the potential immune effects of nitrated food allergens, we investigated the immunostimulatory capacity of our nitrated model allergen BLGn in vitro and in vivo.

In spleen cells of naive, untreated mice, we observed significant differences in the innate cytokine response after stimulation with BLGu or BLGn. This is in contrast with previous investigations using nitrated OVA or nitrated birch pollen allergen Bet v 1 [19] and may be attributed to the characteristics of individual proteins and their amino acid composition. In line with our here presented findings, we have previously reported an enhanced stimulatory potential of BLGn in an allergic organism, but not for nitrated ovomucoid [21].

In the present study, we focused on the implications of repeated gavages of nitrated allergens to mimic regular exposure with a nitrated diet. Gavage of BLGn to non-allergic mice was associated with a reduced capacity to form specific antibodies. In accordance with previous findings [4], nitration did not enhance the risk for sensitization as indicated by a lack of specific IgE in serum as well as in intestinal lavages. While repeated oral exposure to BLGu induced a significant elevation of BLG-specific serum IgA, gavage of BLGn did not influence specific IgA titers. This observation might highlight the decreased resistance of BLGn to digestive enzymes in the stomach. The same susceptibility has already been demonstrated for nitrated OVA, which is degraded within a few minutes in simulated gastric fluid [4]. The capacity of food allergens to induce primary sensitization has been associated with stability against gastrointestinal enzymes [22]. Indeed, proteases, lipases, and hydrolases of the gastrointestinal tract are important gatekeepers with regards to allergenic food proteins [41,42]. Proteolytic activity is enabled by low gastric pH facilitating efficient degradation of proteins [23]. As tyrosine residues are one of the favored target sites of pepsin [43], nitration might even improve access to cleavage sites. Therefore, the bioavailability of intact BLGn at immune induction sites of the intestine is assumably decreased, preventing initial antibody formation as the integrity of structural motifs is not provided. Peptides, however, have been discovered to harbor tolerogenic properties [44,45,46]. In accordance, we discovered significant differences in the cytokine response between mice receiving BLGu and BLGn in re-stimulation experiments of spleen cells. Of note, we observed significantly higher IFN-γ and IL-10 production upon in vitro stimulation of spleen cells derived from the group receiving BLGn compared to BLGu. Increased expression of both cytokines has been reported in response to oral immunotherapy aiming at favorable modulation of an allergic response [47]. The absence of clinically relevant concentrations of mMCP-1 complements our findings in non-allergic mice.

In contrast, we observed significantly higher levels of the anaphylaxis marker mMCP-1 in allergic mice after repeated gavages of BLGn compared to BLGu, although the nitrated formulation was more susceptible to gastric enzymes. Especially in milk allergy, IgE antibodies directed against linear epitopes are associated with persistence of allergy throughout adulthood preventing tolerance induction [48]. Digested BLG could therefore still closely resemble such antigenic structures triggering an allergic response [48,49] which, however, did not result in a drop of core body temperature or in clinical signs of anaphylaxis, which has been previously reported after systemic exposure in the effector phase of food allergy [21]. Furthermore, we could not detect mMCP-1 in allergic mice fed with BLGu. We ruled out blocking by BLG-specific IgA antibodies on the intestinal level as IgA titers were significantly higher in lavages of BLGn mice. Furthermore, we cannot assume the suppression of the immune response by Tregs. As both groups revealed similar distributions referring to levels of specific IgA and IgE in serum after sensitization, we analyzed specific IgG1 in serum which was reported to suppress IgE-mediated systemic anaphylaxis [50]. Indeed, we observed a significant increase after three weekly gavages in mice receiving BLGu but a decline in five of six mice fed with BLGn, suggesting early induction of tolerance. Repeated gavages of an allergen without any adjuvant is expected to result in tolerogenic presentation of antigen. In favor of this, also the rather low levels of mMCP-1 and lack of clinical response in BLGn-fed mice indicate that repeated gavage of allergen can modulate an immune response in previously allergic mice.

## 5. Conclusions

Despite the known limitations of in vivo mouse models regarding their comparability to human immune responses [51], our data propose that nitrated food allergens can change and intercept with immune mechanisms. We reveal BLGn to be associated with a reduced capacity of de novo antibody formation, probably due to its enhanced susceptibility to gastric digestion, decreasing its immunogenicity via the oral route. Moreover, T-cell responses indicated that repeated gavage of BLGn changed the cytokine milieu in previously non-allergic mice. Food per se constitutes a daily challenge for the intestinal immune system and the role 3-NT might play remains yet to be in detail determined. Our findings indicate consequences on the immune response to the ingestion of modified food proteins in non-allergic and allergic individuals.

## Figures and Tables

**Figure 1 nutrients-11-02463-f001:**
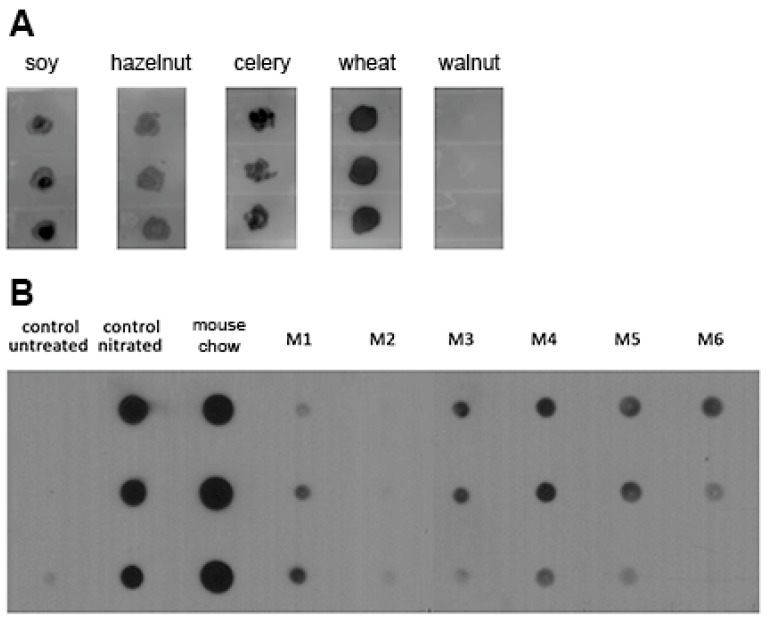
Nitrated tyrosine residues are present in food extracts and stomach content. (**A**) In protein extracts from soy (1 µg), hazelnut (1 µg), celery (1 µg), and wheat (0.33 µg), nitrated tyrosine could be detected by an anti-3-NT antibody in dot blot experiments assaying triplicates. Walnut extract (1 µg) was negative for 3-NT. (**B**) Proteins extracted from the stomach contents of six naïve mice (14 µg) were screened for nitrated tyrosine residues. In five of six mice nitrated tyrosine residues could be detected (M1, M3, M4, M5, and M6) whereas M2 remained negative as compared to the controls.

**Figure 2 nutrients-11-02463-f002:**
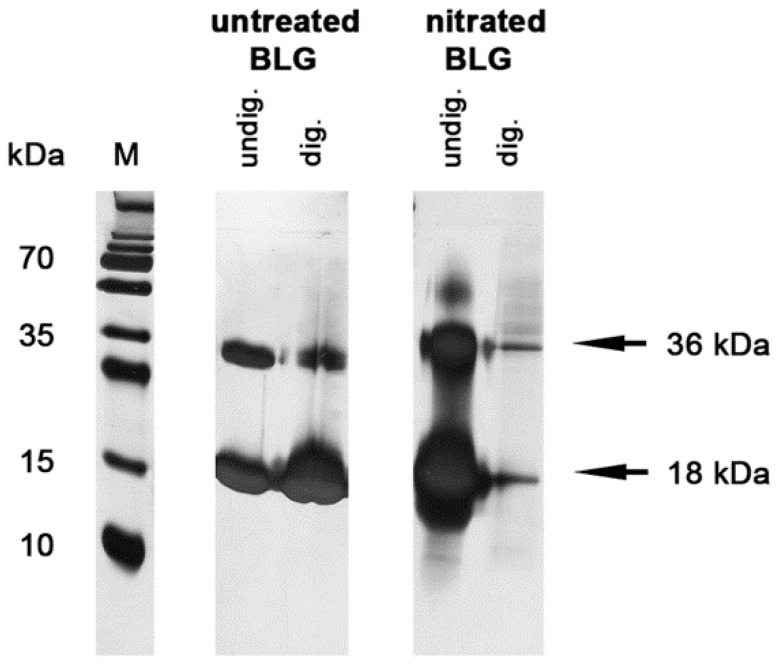
Nitration decreases gastric stability. BLGu (**left panel**) and BLGn (**right panel**) were subjected to gastric digestion for 120 min, and the remaining fragments were visualized in SDS-PAGE. The presence of nitrated tyrosine residues in BLG was discovered to accelerate degradation compared to untreated controls. BLG, beta-lactoglobulin; BLGn, nitrated BLG; BLGu, untreated BLG; M, marker lane.

**Figure 3 nutrients-11-02463-f003:**
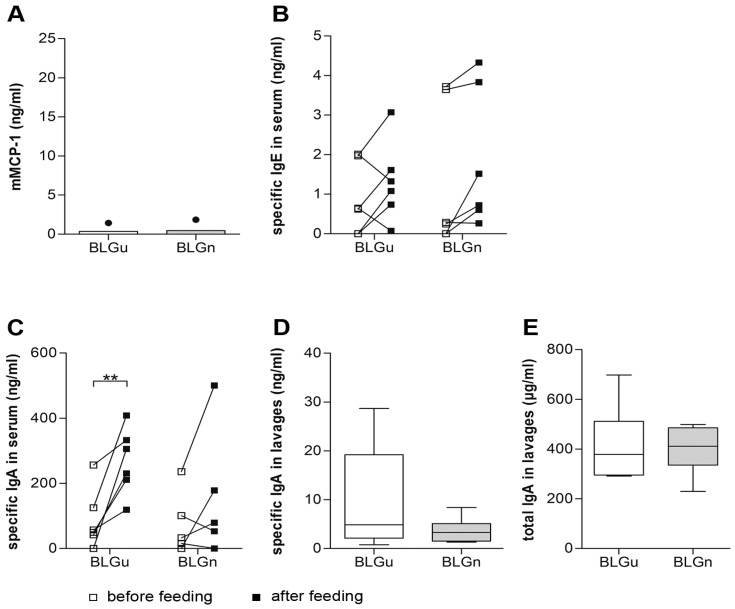
Immune response to oral gavages in non-allergic mice. (**A**) Serum collected 1 h after the last gavage did not show significant levels of mMCP-1. (**B**) Serum samples collected before and after oral gavages of BLGu and BLGn were analyzed. Neither BLGu (paired *t*-test) nor BLGn (Wilcoxon signed-rank test) induced significant production of IgE in the serum of previously non-allergic mice after three oral gavages. (**C**) BLGu elicited a significant elevation of specific IgA in serum (paired *t*-test). (**D**) Differences of specific IgA in lavages did not reach statistical significance between groups (unpaired *t*-test with Welch correction) (**E**) Total IgA in lavages did not differ comparing both groups after gavages (unpaired *t*-test). ** *p* < 0.01; BLG, beta-lactoglobulin; BLGn, nitrated BLG; BLGu, untreated BLG, mMCP-1 mouse mast cell protease-1.

**Figure 4 nutrients-11-02463-f004:**
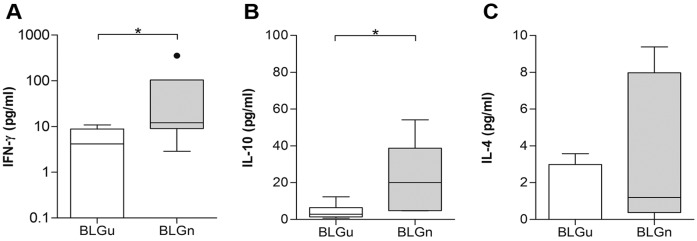
Re-stimulation of spleen cells with BLGu indicates the influence of treatment on cytokine production in non-allergic mice. Isolated spleen cells from non-allergic mice fed with BLGu or BLGn were stimulated with BLGu to evaluate the cytokine production upon allergen encounter. Baseline levels were subtracted. Spleen cells of BLGn-gavaged mice produced significantly higher levels of (**A**) IFN-γ (Mann–Whitney-U-test) and (**B**) IL-10 (unpaired *t*-test) compared to group BLGu. (**C**) IL-4 remained mostly below the detection limit of the assay used and did not show any differences (Mann–Whitney-U-test). * *p* < 0.05; BLG, beta-lactoglobulin; BLGn, nitrated BLG; BLGu, untreated BLG.

**Figure 5 nutrients-11-02463-f005:**
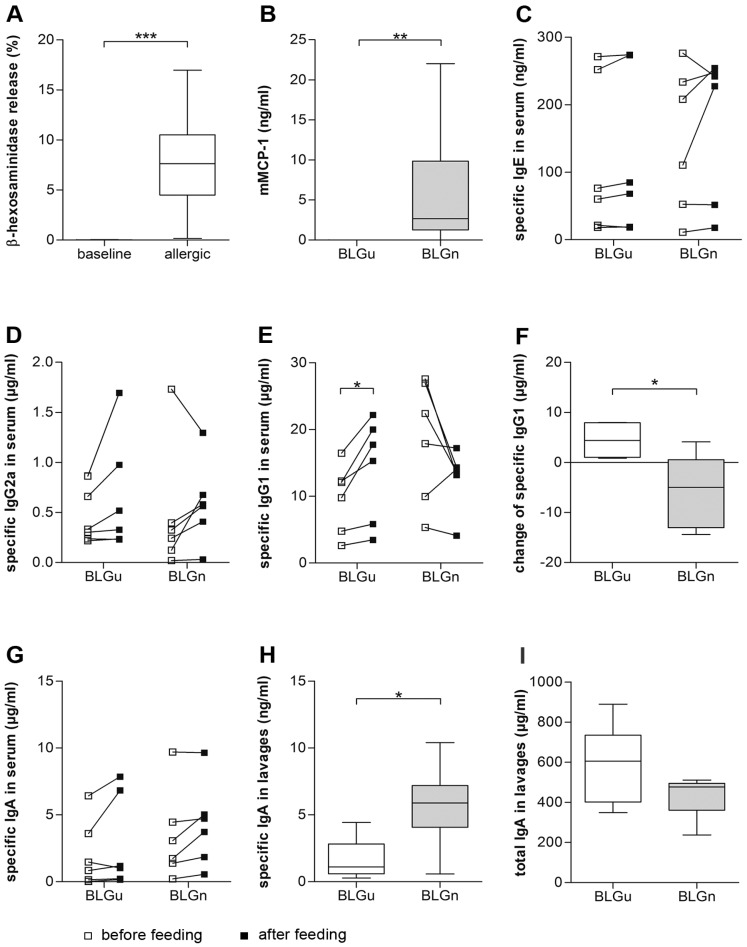
Immune response to gavages in allergic mice. Serum samples collected after sensitization with BLGu and after three oral gavages were analyzed. After randomization into the two experimental groups, baseline values (white squares) did not differ. (BLG specific IgE (unpaired *t*-test, *p* = 0.63), IgA (unpaired *t*-test, *p* = 0.45), IgG2a (Mann–Whitney-U-test, *p* = 0.67), and IgG1 (unpaired *t*-test, *p* = 0.07)). (**A**) Immunization induced BLG-specific IgE titers capable of eliciting mediator release from RBL-cells when stimulated with BLGu. (**B**) Allergic mice fed with BLGn exhibited a significantly higher mast cell-related response than mice fed with BLGu (Mann–Whitney-U-test). (**C**) Gavage of BLGu (paired *t*-test) or BLGn (Wilcoxon signed-rank test) did not significantly enhance specific IgE production. (**D**) Changes of specific IgG2a were not significant in response to BLGu (paired *t*-test) or BLGn (Wilcoxon signed-rank test). (**E**) Gavage of BLGu induced significant upregulation of specific IgG1 (paired *t*-test). (**F**) Alterations of IgG1 were significantly different comparing both BLG preparations (unpaired *t*-test). (**G**) Specific IgA levels in serum were unaltered by gavages of BLGu (Wilcoxon signed-rank test) and BLGn (paired *t*-test). (**H**) Mice receiving BLGn had ultimately higher specific IgA levels in intestinal lavages but (**I**) comparable titers of total IgA. * *p* < 0.05; BLG, beta-lactoglobulin; BLGn, nitrated BLG; BLGu, untreated BLG.

**Figure 6 nutrients-11-02463-f006:**
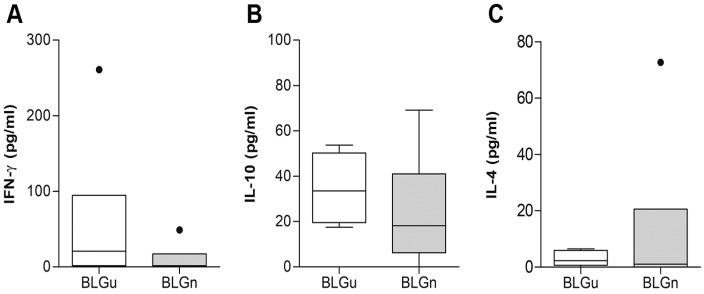
Re-stimulation of spleen cells with BLGu does not indicate an influence of treatment on cytokine production in allergic mice. Isolated spleen cells of BLGu-allergic mice fed with BLGu or BLGn were stimulated with BLGu, and supernatants were screened for the presence of cytokines. No significant differences between BLG preparations were detected after baseline subtraction for (**A**) IFN-γ (Mann–Whitney-U-test), (**B**) IL-10 (unpaired *t*-test), (**C**) or IL-4 (unpaired *t*-test with Welch correction). BLG, beta-lactoglobulin; BLGn, nitrated BLG; BLGu, untreated BLG.

**Table 1 nutrients-11-02463-t001:** In vitro cytokine production of spleen cells from naïve animals in response to BLGu and BLGn.

Cytokine	BLGu Median (IQR)	BLGn Median (IQR])	*p* Value
IFN-γ (pg/mL)	59.65 (26.90–201.00)	442.40 (128.00–781.50)	0.03
IL-10 (pg/mL)	0.00 (0.00–8.59)	0.00 (0.00–3.38)	0.60
IL-4 (pg/mL)	8.75 (5.86–16.19)	4.13 (1.86–23.81)	0.63

Cytokine levels after BLGu and BLGn stimulation were compared by Mann–Whitney-U*-*test. IFN-γ was significantly enhanced by stimulation of BLGn compared to BLGu. BLG, beta-lactoglobulin; BLGn, nitrated BLG; BLGu, untreated BLG; IQR, interquartile range.

**Table 2 nutrients-11-02463-t002:** Relative and absolute Treg counts from spleens and pooled lymph node cells of non-allergic or allergic animals after gavages of BLGu or BLGn.

Immune Status	Gavage	Treg Count (Pooled Lymph Node Cells)	Treg Count (Spleens; mean ± SEM)	*p* Value	Tregs (%; Pooled Lymph Node Cells)	Tregs (%; Spleens; mean ± SEM)	*p* Value
non-allergic	BLGu	495	382.0 ± 15.0	0.47	5.06	10.8 ± 0.46	0.23
	BLGn	529	369.5 ± 6.90		6.59	11.5 ± 0.34	
allergic	BLGu	474	319.3 ± 21.6	0.56	8.29	9.76 ± 1.05	0.40
	BLGn	677	333.5 ± 23.2		10.1	10.8 ± 0.57	

Percentage of Tregs (given as Tregs/CD4^+^ cells) and total Treg counts are compared using unpaired *t*-tests between experimental groups. Data are presented as mean ± SEM. BLG, beta-lactoglobulin; BLGn, nitrated BLG; BLGu, untreated BLG; SEM, standard error of the mean.

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
