# Peer review of "Immune Effects of the Nitrated Food Allergen Beta-Lactoglobulin in an Experimental Food Allergy Model"

_nutrients, 2019, doi:10.3390/nu11102463_

Round 1
Reviewer 1 Report
The manuscript by Ondracek et al., entitled “Immune effects of the nitrated food allergen beta3 lactoglobulin in an experimental food allergy model” describes the potential effect of the ingestion of nitrated food proteins on the immune response. The authors first show the presence of nitrated proteins in food and mouse stomach content. Using the model milk protein, beta-lactoglobulin (BLG), that is chemically nitrated they next illustrate differences in production of some cytokines and immunoglobulins by mice splenocytes and serum/intestinal lavages, respectively. Although the manuscript is interesting the data supporting the conclusions are superficial. There are several issues in the manuscript and only some are mentioned by us. For instance, splenocytes are studied as correlates of lamina propria cells of small intestine or colon, these results should be present in the manuscript (even mesenteric lymph nodes might have been more relevant). Next the authors provide IgA content in the serum and gut intestinal lavage yet only show IgG levels in the serum; luminal IgG levels are not shown although they might also contribute to the allergic response. There seems to be discordance in overall hypothesis: if BLGu, being more stable, induces greater immune response than BLGn (Fig 3) in naïve mice, what is the role of BLGn (especially that the mice are sensitized with BLGu)? The data for splenic T regulatory cells is meaningless: 1. absolute numbers need to be provided not only percentages, 2. Functional assessment of these cells by performing suppression assay would be more informative, 3. Technically speaking, how were dead cells excluded from the analysis (FSC and SSC based gating is not adequate)? Why were the splenocytes incubated with 2.5 ug/ml of BLGu/BLGn specifically and not with titrated amounts? Total IgA serum levels are usually in the range of IgG1 or IgG2a levels, why where the dilutions greater for IgA?
Reviewer 2 Report
“Immune effects of the nitrated food allergen beta- lactoglobulin in an experimental food allergy model” looks interesting and in area of interest part of the group. The Authors study the other food antigen OVA very similar way but it is acceptable (Nutrients, PLOS ONE).
In my opinion the manuscript needs some corrections to improve its quality and clarify the experiments.
Major comments to whole manuscript:
The term feeding used in all manuscript, suggest that it is the main source of protein in the diet – what was not the subject of presented experiments. Please take changing the word for treatment or gavage into consideration. The term naïve mice used by whole text is not proper. Mice are naïve just until first contact with antigen. Conclusions have to be rewritten. BLG is the second after casein cow milk allergen; it is also quite resistant to digestion. There is lots of literature about this. Fact that modification changed its immune properties of course exist the question is only in which direction – decrease its immunogenicity or not? Reference: remove double numberingMinor comments:
Line 24: “repeatedly ingested 
…. To naïve mice”
I would like to say that mice are naïve until first treatment what means day 0, and then they are not naïve any more. I do not find "repeatedly ingestion" as the aim of the study. Please clarify that.
Line 27: “food groups” experimental groups will be more proper 

Line 64: There is scoring system for anaphylactic shock based on visible mice reaction, it should be noted about this – yours reference 26
Line 71-74: I am still not sure which experimental group was naïve? As I wrote above mice are naïve until first contact with antigen. Even Group 1 or 2 are not finally naïve because BLGu or BLGn were gavage to the mice. And why those group did not get PBS in the same schedule as group 3 and 4 to eliminate stress parameter.
Line 75: Reagents.
I propose split this mixture of all reagents for the appropriate method. It will be clearer. It is enough to put manufacturer, city and country after the reagent name.
Line 118: proteins are isolated
Line 154—177:
Please rewrite this. Omit repeating “Clinical….” (Line 159-160 and 173-174).
Cycle is something what has more than one procedure. In this sense immunization this is not a cycle. Immunization by the way is the term, which concern on moment when antibodies producing starting, later there is boosting. Please correct it.
In the immunization and antigen treatment description, please provide groups names.
Line 185: cytokine concentration
Line 193: phenotyped will be more scientific decryption
Line 205: Gating strategy on supporting information 2.
Usually we gate lymphocytes on FSC vs. SSC in the range 150-200, then CD4 vs. CD25 to gate CD4+CD25+ and that from the last one FoxP3+. Author’s strategy suggests that in “cells” population we have more than lymphocytes. CD4 marker exists also on Macrophages and Monocytes and Granulocytes. CD25 exist also on B cells, NK cells. Why Authors do not gate CD4 vs. CD25 to get clear population CD4+CD25+. Culturing cells of course can shift the cells on forward scatter but this is not clear in this form. Please recheck the results.
Round 2
Reviewer 1 Report
The authors have replied to all of my concerns.